# Hybrid Vigour and Hybrid Mimics in *Japonica* Rice

You Zhang [1,2], Ben Ovenden [3], Yao He [4], Wenwei Ye [4], Xianjun Wu [4], William James Peacock [1,2,*]
and Elizabeth S. Dennis [1,2,*]

1   Faculty of Science, University of Technology Sydney, Broadway, Sydney, NSW 2007, Australia;
    zhangyoumax@163.com
2   CSIRO Agriculture and Food, Canberra, ACT 2601, Australia
3   NSW Department of Primary Industries, Wagga Wagga Agricultural Institute,
    Wagga Wagga, NSW 2650, Australia; ben.ovenden@dpi.nsw.gov.au
4   Key Laboratory of Southwest Crop Genetic Resources and Genetic Improvement, Rice Research Institute,
    Sichuan Agricultural University, Ministry of Education, Wenjiang, Chengdu 611130, China;
    heyao2017@foxmail.com (Y.H.); yewenwei0723@163.com (W.Y.); wuxjsau@126.com (X.W.)
*   Correspondence: jim.peacock@csiro.au (W.J.P.); liz.dennis@csiro.au (E.S.D.)

**Abstract:** In crop improvement programs, hybrid vigour (heterosis) is an important breeding strategy but the molecular mechanisms of hybrid vigour are still unclear. Grain yield declines after F1 generation due to phenotypic segregation. We found that, at the early seedling stage in hybrids derived from the temperate *japonica* rice varieties 'Doongara' and 'Reiziq', hybrid vigour was approximately 40% greater than in the better parents. Inbred high-yielding lines (Hybrid Mimics) were developed from the 'Doongara' × 'Reiziq' F1 by selfing and recurrent selection for F1-like plants in the F2 through to the F5 generation. Grain yields are stable over subsequent generations in the Hybrid Mimic lines. The importance of photosynthesis in early seedling development was demonstrated. Photosynthesis-related genes were expressed in the hybrid earlier than in the parents; physiological evidence using gas exchange indicated the early commencement of photosynthesis. Dark germination experiments supported the requirement for photosynthesis for early vigour in hybrids.

**Keywords:** heterosis; hybrid mimics; photosynthesis

## 1. Introduction

Hybrid vigour, or heterosis, is the phenomenon wherein hybrid offspring have seed yield and plant growth greater than that of their parents [1]. Heterosis is a breeding strategy that has been used in some crops for decades. Although hybrid vigour enhances yields in crop species such as maize, *indica* rice, canola and sorghum [2], the molecular mechanisms underpinning hybrid vigour are still elusive [3]. Two genetic hypotheses were initially suggested, namely the dominance hypothesis [4] and the overdominance hypothesis [5,6]. Quantitative trait locus (QTL) studies have supported different genetic models—dominance [7,8], overdominance [9–11], pseudo-overdominance [12,13] and epistasis [10,11,14]—indicating that the molecular basis of heterosis results from the action of many genes. In all hybrids, the level of hybrid vigour in the F1 generation declined in subsequent generations [15]; the F2 generation shows segregation for morphological, developmental and seed yield traits. The phenotypic variation in the F2 generation makes it undesirable for farmers to use the seeds of the F1 hybrid for the next planting season.

Rice (*Oryza sativa* L.), a widely consumed staple food, consists of two subspecies: *indica* and *japonica* [16]. Rice has a high self-pollination rate with a natural out-crossing frequency up to 5% [17]. The high rate of self-pollination is due to the physiological and anatomical structure of the plant, which makes producing commercial rice F1 hybrid seeds by physical emasculation impractical [18]. Male sterility and fertility restoration systems have provided the opportunity for the large-scale commercial production of F1 hybrid rice seeds [19]. In *indica* rice, the current male sterility and fertility restoration systems include three-line and

two-line male sterile systems. The three-line system uses a male sterile line, a restorer line and a maintainer line [20]. In the two-line system, the fertility of the male-sterile line is photo-temperature sensitive, which means that the fertility of the sterile line can recover under certain temperature and photoperiod environmental conditions [20]. In China, the establishment of male sterile *indica* hybrid systems enhanced the commercial rice yield nearly two-fold between 1976 and 1995 with hybrids having up to 30% increased yield over inbred lines [18,21]. Hybrid *indica* rice has been widely adopted and comprises over half of the rice planting area in China [22], while hybrid *japonica* varieties are grown over less than 3% of the planting area [23].

In Arabidopsis, we developed a method that fixes the agronomic traits of the F1 hybrid through a recurrent selfing and selection system to produce stable, pure breeding, and high yielding lines without the need for male sterile lines. Elite individuals were selected in each generation and selfed to produce the next generation. At the F5 generation, large plant phenotypes (Hybrid Mimics) were similar to the F1 hybrid. The F1-like phenotype was inherited to the F6 generation and beyond. After six generations, chromosome segmental sequences were mostly homozygous with segments derived from both parents [15].

Hybrid mimic selection in *indica* rice varieties produced mimic lines from two Chinese hybrids, 'FLY1' ('GZ63S'/'93–11') [24] and 'DY527' ('D62B'/'R527') [25]. Both lines have close to 40% yield gain compared to their parents. The Hybrid Mimic lines contain genomic segments from both parents and by the F6 generation, are mostly homozygous for one parental allele at each locus [26]. The high level of homozygosity of the Mimics ensures that the F1-like phenotypes are fixed; the segregation of different plant morphologies is insignificant in the offspring of the Mimics.

Based on the experience of Mimic selection in the two *indica* hybrids [26], we aimed to develop Hybrid Mimic lines in the other rice subspecies, *japonica*. We obtained F5 Mimic lines with yield results not significantly different to the F1 hybrid and approximately 40% greater than the better parent. The phenotypic heterogeneity of the plants in the Mimic lines was decreased compared to the heterogeneity in the F2 generation.

Previous studies indicated that both hybrids and Hybrid Mimics have early vigour compared to the parents and express photosynthetic genes earlier than the parents, maintaining the growth vigour throughout the life cycle [26,27]; this result is limited to transcriptional evidence. Our study collected physiological evidence of the early initiation of photosynthesis in hybrids using Q2 gas-exchange technology [28]. We also showed germination and early growth in the dark abolished hybrid vigour. Our studies demonstrated the necessity of photosynthesis for hybrid vigour establishment at the seedling and later stages.

## 2. Materials and Methods

Rice cultivation and breeding in Australia is based on inbred *japonica* long grain and medium grain varieties. A phylogenetic analysis was conducted across parental varieties and selected breeding lines used in the Australian rice breeding program. Whole genome profiles of DArTseq markers were generated for each line using the genotyping procedure given in [29]. The resulting set of 42,652 markers was used to perform a hierarchical cluster analysis using Ward's criterion [30]. Cluster analysis output was used to construct a phylogenetic tree using the package ape [31] in the R environment [32]. The phylogenetic analysis indicated the presence of two genetically distinct elite germplasm pools generally corresponding to the long and medium grain breeding programs (Figure 1). In order to exploit possible heterosis across the two germplasm pools, hybrids were generated using representative parents from each germplasm pool: 'Doongara', a semi-dwarf *japonica* long grain variety released in 1989 [33], and 'Reiziq', a semi-dwarf *japonica* medium grain variety released in 2005 [34].

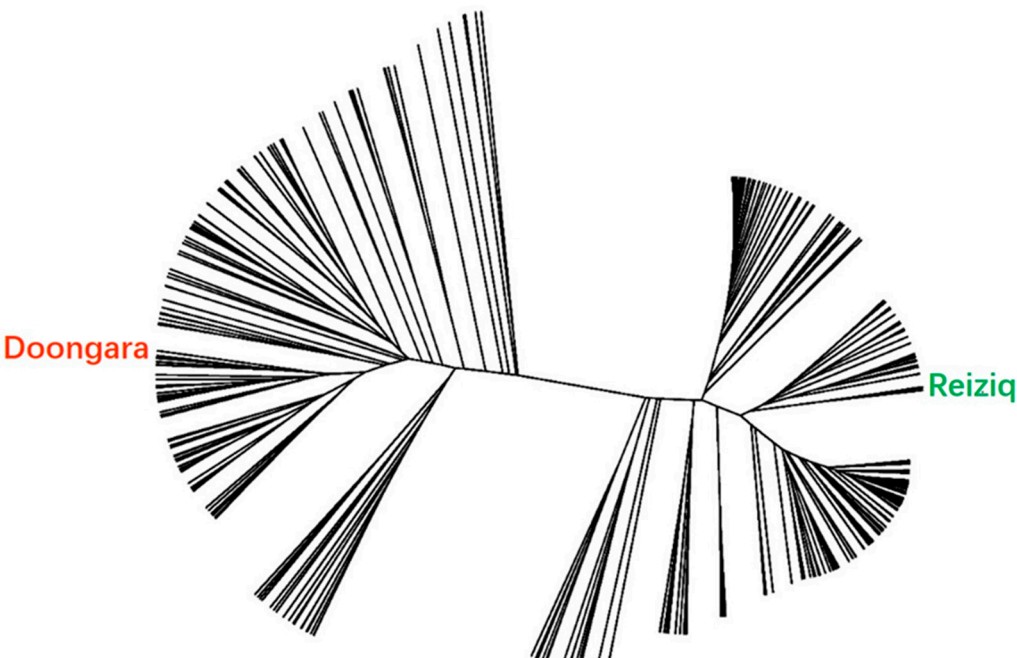

**Figure 1.** Phylogenetic tree of 'Doongara' and 'Reiziq'. The two main clades represent the division in the breeding program between the *japonica* long grain varieties (represented by 'Doongara') and *japonica* medium or short grain varieties (represented by 'Reiziq').

The Australian varieties 'Doongara' and 'Reiziq' were selected as the parents for the hybrid mimic program (Figure 1). F1 seeds were germinated on wet filter paper and incubated at 30/25 °C (30 °C during daytime and 25 °C at night) for seven days. One germinated seedling was transplanted into each pot (175 mm tall, 175 mm diameter, 2.8 L volume) containing local topsoil (Birganbigal clay loam) [35] in a glasshouse at Yanco Agricultural Institute, Yanco, Australia (elevation:138 m, location 34°37′16.88″ S, 146°24′43.79″ E). Pots were arrayed in a randomized complete block design with 10 replicates for each genotype ('Doongara', 'Reiziq' and F1) and three sowing times for a total of 90 pots. The placement of genotypes within the experimental layout was optimized with the spatial design package DiGGer [36]. The first sowing was planted in the glasshouse on 10 November 2017, and the second and third sowings were planted on 21 November and 28 November, respectively, so that the growth coincided with the rice growing season in southern Australia. Soil pots were placed in water tanks and submerged to a depth of 1–3 cm over the soil surface. The temperature was controlled at 25 °C. Air temperature in the glasshouse was controlled at 25 °C (night) and 30 °C (day). Plant available radiation was measured with a SunScan type SS1 (Delta-T Devices, Cambridge UK) at a canopy height at 10:00 a.m. 40 days after each sowing date, within a range of 824.3–1323.3 mM PAW observed over the experiment compared to 1309.7–1950.0 mM outside the glasshouse.

Leaf width and length were recorded 20 days after sowing and multiplied by 0.7 [37] to calculate leaf area. Tiller number was recorded at 40 days after sowing. The height of each tiller to the panicle collar was measured in mm. The above-ground section was cut-off to avoid the loss of dry leaves or seeds. Each individual panicle was stripped into a separate envelope to compute the number of panicles per plant. Harvested samples were placed in an air-conditioned room for 10 days before weighing. The above-ground section of the rice plants was harvested to measure the biomass of each plant. The grain yield was measured by weighing the threshed grains of each plant. Biomass weight was measured at physiological maturity (17 March 2018). Phenotypic data were analysed using the mixed models software package ASREML-R version 3 [38] in the R environment [32], and Best Linear Unbiased Estimates were obtained for each trait.

- Selection for hybrid mimics

The F2 population of the 'Doongara' × 'Reiziq' hybrid contained 324 individuals which were sown on 3 July 2018. Five F1-like F2 plants and one small control F2 plant were selected for further growth. The selection was based on individual plant grain yield and biomass. The seeds from the selected F2 and the small control plant were used to develop the F3 generation (Supplementary Table S2). Forty F3 seeds from each F2 plant were sown as F3 plants (on 20 March 2019). In the F3 generation, four elite F3 individuals were selected based on grain yield and biomass and were used to generate the F4 Hybrid Mimic generation and the small control line. A total of 165 individuals were sown in the F4 generation, which included 15 replicates of 'Doongara', 'Reiziq', the F1 hybrids and the four F4 lines and their F3 parental lines on 6 November 2019. From the F2 to F4 generation, the seeds were germinated on wet filter paper and incubated at 30/25 °C with F1 individuals as controls. These generations were planted and grown in the glasshouse at the CSIRO Black Mountain Laboratories, Australia (location 35°15′43.7″ S 149°06′19.1″ E). The planting soil contained 1/3 Debco Garden Mix (Bella Vista, NSW 2153, Australia) and 2/3 CSIRO rice potting mix. The placement of genotypes within the experimental layout was optimized with the spatial design package DiGGer [36]. The growing conditions and crop arrangement were similar to F1 planting in Yanco, but the soil pots used were smaller (120 mm tall, 150 mm diameter, 1.2 L volume) to provide more individuals for selection by increasing the planting density.

- Field evaluation

The parental lines and the seed from the F1 to the F5 lines were planted in the field in Wenjiang, Sichuan, China, on 12 May 2020 (location 30°67′46.0″ N 104°06′53.9″ E) with 20 plants in each line. Seeds were soaked in water for 2–3 days and then sown into the seedling bed field. The whole seedling bed field used a plastic film surrounded by a ditch to sustain a suitable growth temperature for seedlings. The water level of the ditch was increased to a depth of 1–3 cm on the soil surface 7 days after sowing (DAS). At 30 DAS, seedlings were transplanted into the paddy field spaced at 15 cm for each line. The distance between the two different lines was 35 cm. During the growing season, there was 800–1600 mm precipitation and 7000–1000 h of sunshine. The biomass and grain yield of each individual were measured by weighing the above-ground section and the threshed grains of each plant. For growth and physiological parameters, statistical analysis was Dunnett's one-way ANOVA which was performed using GraphPad Prism (v 8) software.

- Real-time qPCR

RNA of 'Doongara' and 'Reiziq' and the F1 hybrid was extracted from the plant coleoptile tissue by use of the Maxwell® RSC Plant RNA Kit and the Maxwell® RSC instrument (Promega, Fitchburg, WI, USA). RNA was reverse-transcribed to cDNA by a SuperScript III First-Strand cDNA Synthesis Kit as well as RNaseOUTTM Recombinant Ribonuclease Inhibitor (Invitrogen, Carlsbad, CA, USA) for real-time qPCR assessment. Ten photosynthetic genes were chosen as the target genes for RT-qPCR (Supplementary Table S4) and Actin was used as the housekeeper gene. The raw fluorescence data of qPCR were analysed by LinRegPCR software [39,40]. The expression data were normalized against the housekeeper gene, actin, and calculated as fold changes against MPV.

- Q2 gas-exchange photosynthetic measurement

The Q2 $O_2$-sensor system (Astec Global, Maarssen, The Netherlands) is designed and marketed for seed germination assays and is used to obtain automated, high-throughput fluorophore measurements for oxygen consumption. The net photosynthesis rate was measured by oxygen production with additional illumination provided. The Q2 experiments used 4 mL tubes to hold a single seed with 600 µL distilled water. Light with 250 µmol·s$^{-1}$·m$^{-2}$ intensity was provided during the 72 h measurement. The rice seeds of 'Doongara', 'Reiziq' and the F1 hybrid (5 replicates) were pre-incubated at 48 °C under dry conditions and then directly sown in 4 mL tubes. To avoid an excessively high $CO_2$

concentration inside the tubes, the oxygen level was checked every 6 h. If the oxygen level was lower than or close to 80% of air level, the gas was exchanged by opening the tubes. The experiments were repeated. Oxygen consumption was calculated from the oxygen change curve of each sample, and the natural logarithm of the oxygen consumption curve was plotted. A linear trend line was plotted based on the natural logarithm of the oxygen consumption curve of the first dozen hours (when the natural logarithm oxygen consumption curve is approximately linear). A divergence point, when photosynthesis begins, was identified as the time point when the difference between the natural logarithm value and trend value is larger than 2% of the natural logarithm value.

## 3. Results

### 3.1. Selection of Japonica Rice Hybrids for Development of Hybrid Mimics

The early growth patterns of the selected hybrid lines (bred from 'Doongara' and 'Reiziq') were assayed to select lines with heterosis at the seedling stage. Among these hybrids, the 'Doongara' × 'Reiziq' F1 line showed the highest level of growth vigour at 14 days after sowing (DAS) (Brown–Forsythe *p* value = 0.0015) (Figure 2a and Figure S1). The F1 line had a 40% increase in grain yield at maturity compared to the parents (Brown–Forsythe *p* value = 0.0091) (Figure 2). The hybrid showed increased seedling leaf area (Figure 2c) and tiller number at 40 days after sowing (Figure 2d) compared to the parents. This hybrid was chosen for Hybrid Mimic selection, based on its early growth advantage and hybrid vigour in grain yield.

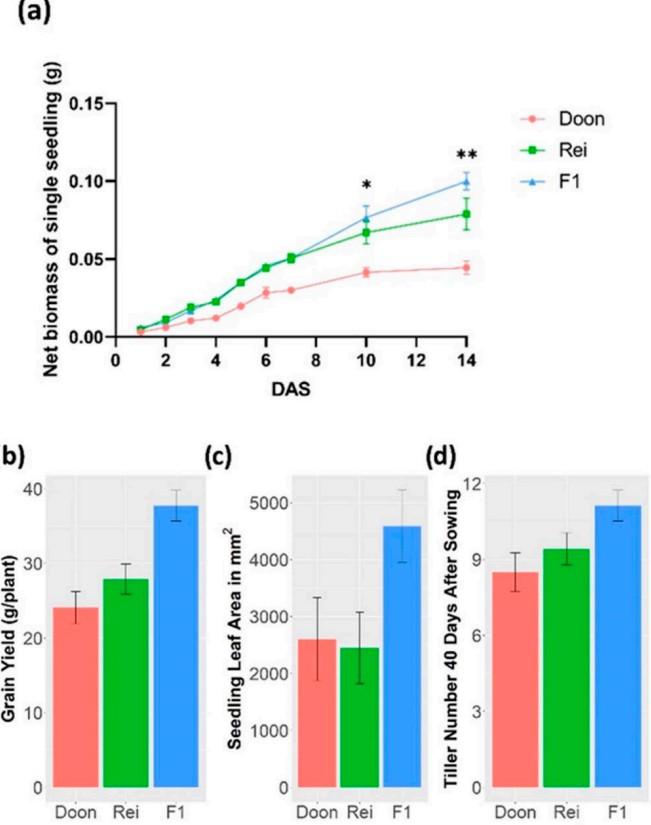

**Figure 2.** Growth parameters of 'Doongara', 'Reiziq' and the F1 hybrid. (**a**) Two-week growth pattern comparison of 'Doongara' × 'Reiziq'; (**b**) Best linear unbiased estimates of harvest grain weight of 'Doongara', 'Reiziq' and the F1 hybrid; (**c**) Best linear unbiased estimates of the average leaf area of the F1, 'Doongara', and 'Reiziq' at Week 3 after sowing; (**d**) Best linear unbiased estimates of tiller number of the F1, 'Doongara', and 'Reiziq' at Day 40 after sowing. * *p*-value < 0.05, ** *p*-value < 0.01. *p*-values were calculated using Dunnett's one-way ANOVA. Error bars represent in (**a**) standard error of the mean (SEM). Error bars in (**b**–**d**) represent 95% confidence intervals.

### 3.2. Hybrid Mimic Selection from a Japonica Rice Hybrid

Hybrid mimic selection from the 'Doongara' × 'Reiziq' hybrid started with an F2 population of 324 individuals under glasshouse conditions. Supplementary Table S1 shows the yield data of 50 individuals from the 324 F2 population, and the segregation of the biomass and grain yield. From the F2 population (Supplementary Table S3), five plants with the greatest grain yield or biomass were chosen to develop the F3 population. Forty seeds from each selected plant were sown as one F3 line and the seed from four F3 plants with the greatest grain yield were grown to produce the F4 generation. Two lines were chosen from the F4 to develop the F5 generation. Earlier generations of plants from the parental generation to the F4 generation were planted together with the F5 generation. The phenotypic variation of the F5 plants in grain yield and biomass was decreased relative to the F2 generation and the grain yield approached that of the F1 hybrid (Figures 3 and 4). Plant height, panicle number and seed setting rate were also measured on the parent to F5 generation set (Supplementary Figure S4). The panicle number and plant height trended to approach the F1 level, but the seed setting rates were different among generations (Supplementary Figure S4).

### 3.3. Heterosis of the F1 Hybrid Started in Early Development

Seeds of the F1 hybrids and parental lines were germinated and biomass measured two weeks after sowing (Figure 2a). At 14 days after sowing, the F1 hybrids showed significantly greater biomass relative to both parents. The greater biomass continued to develop until the grain filling stage. During the vegetative stage, the seedling leaf area and tiller number both showed hybrid vigour (Figure 2c,d and Figure 5). The F1 hybrids had a larger plant size than the parental individuals at 3 weeks after sowing and the plants at the grain filling stage had a greater plant height and panicle number (Figure 5). At harvest, the F1 had a significantly higher grain yield than both parents (Figure 2b).

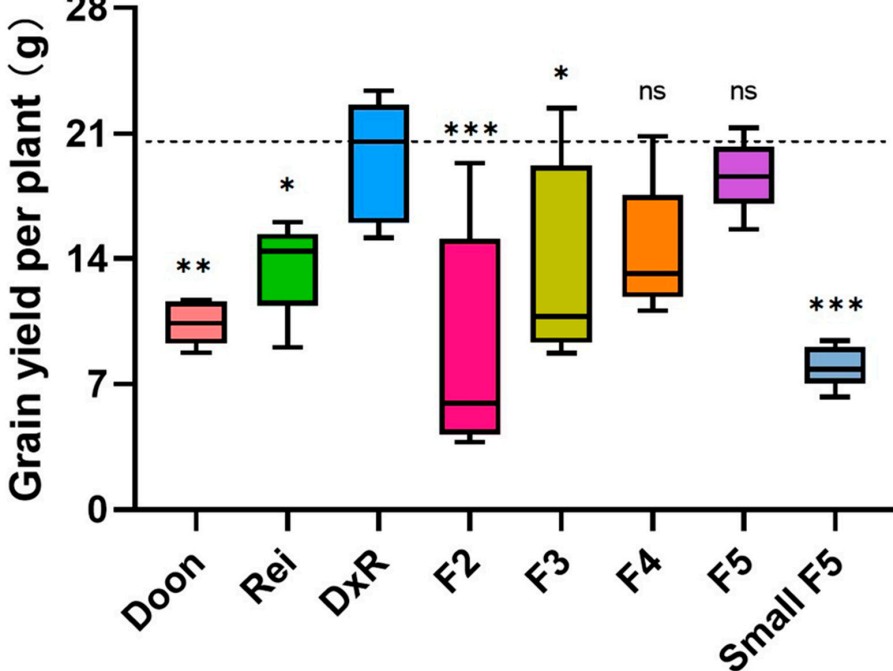

**Figure 3.** Boxplots of grain yield per plant from parental lines to the F5 generation. The graph shows the grain yield changing over successive generations during selfing and selection. The F2 lines have a lower grain yield with larger variation compared to the F1 hybrid. The grain yield in the F5 line is similar to that in the F1 line. * *p*-value < 0.05, ** *p*-value < 0.01, *** *p*-value < 0.001. *p*-values were calculated using Dunnett's one-way ANOVA. Error bars represent the maximum and minimum yield levels.

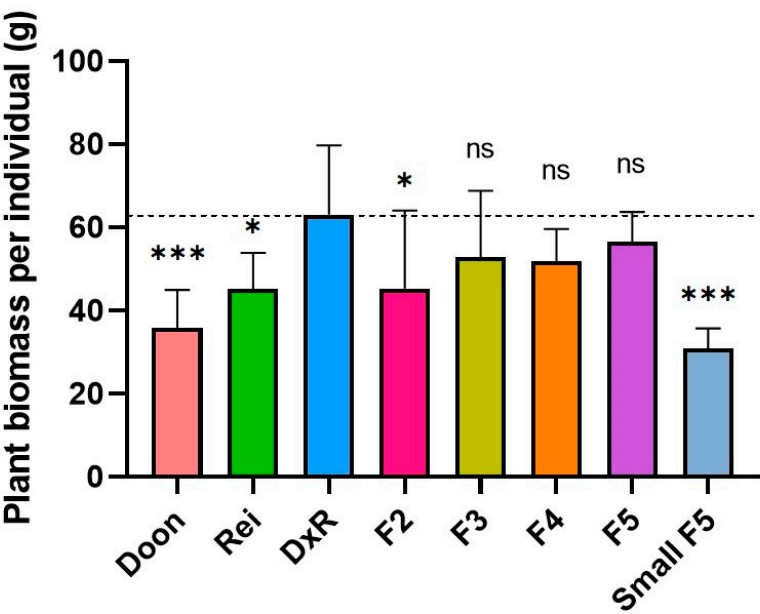

**Figure 4.** The plant biomass per individual plant from parental lines to the F5 generation. The bar chart shows the plant biomass change over successive generations during selfing and selection. F2 lines have a lower biomass with larger variation compared to the F1 hybrid. Grain yield in the F5 line is approaching the F1 line. * *p*-value < 0.05, *** *p*-value < 0.001. *p*-values were calculated using Dunnett's one-way ANOVA. Error bars represent the standard error (SE).

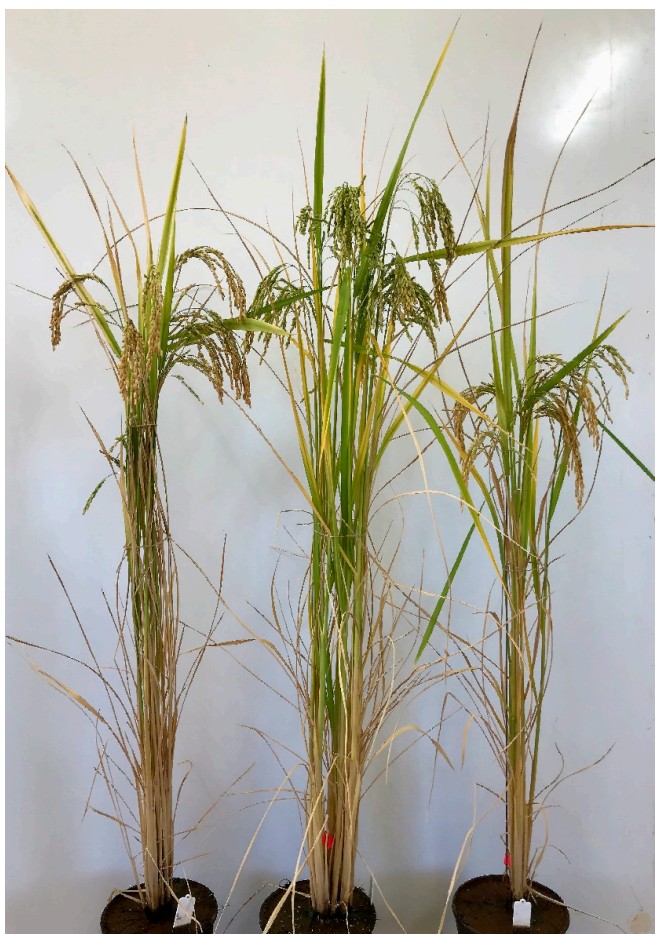

**Figure 5.** Growth comparison of the representative plants of F1, 'Doongara' and 'Reiziq' at the grain filling stage. Left is 'Reiziq', Right is 'Doongara' and in the middle is the F1 hybrid.

The activity of ten photosynthetic genes of germinated seeds and seedlings of parents and the 'Doongara' × 'Reiziq' hybrid were measured at 1, 2 and 3 days after sowing (DAS) (Figure 6). Photosynthesis-related genes were more highly expressed in the F1 hybrid at 1 DAS with similar expression levels to MPV at 2 and 3 DAS (Figure 6). The initial expression of the photosynthetic genes was earlier in the hybrids than in the parental lines. The early vigour resulted in the 'Doongara' × 'Reiziq' hybrid line with a larger total leaf area than the parents; the carbon assimilation rate per unit leaf area was similar [26,36]; because of their larger leaf area, the hybrids produce more total photosynthate than the parental lines, 'Doongara' and 'Reiziq'.

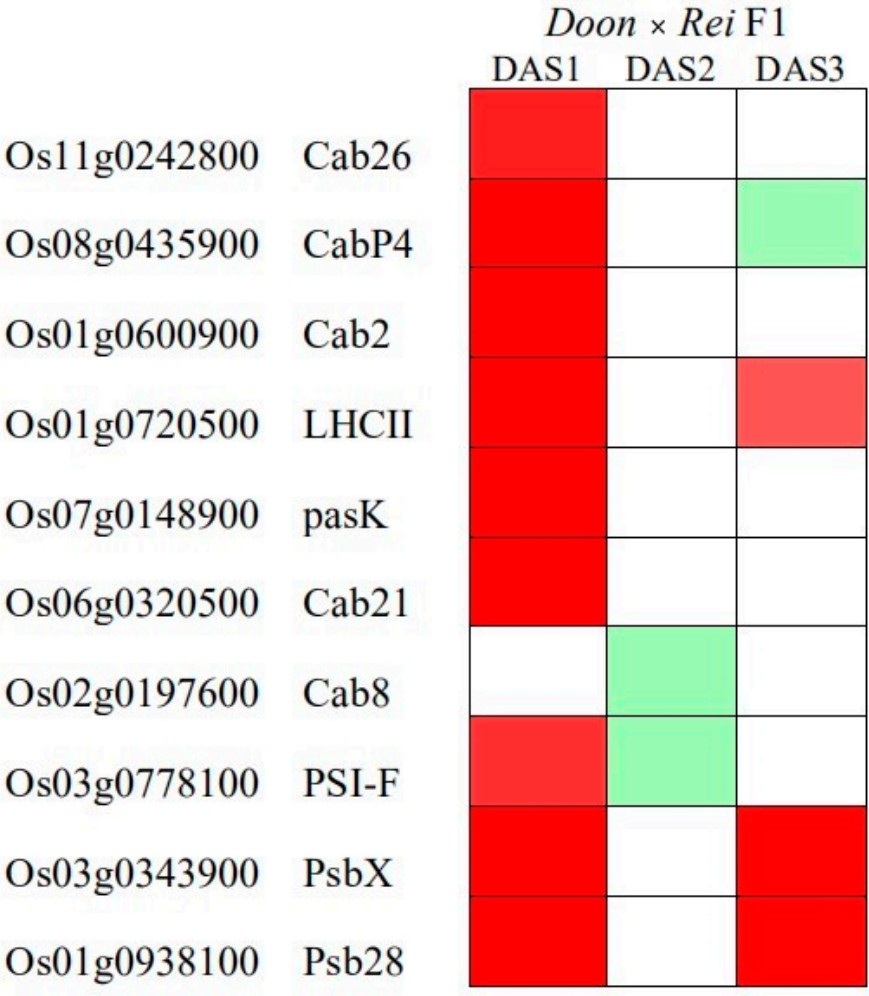

**Figure 6.** Heatmap of expression level foldchange (FC) of photosynthesis-related genes to MPV at different time-points in the Doon × Rei Hybrid (red: FC > 1.3, green: FC < 0.77, white: not differentially expressed). Detailed gene information is available in Supplementary Table S4.

The photosynthetic gas exchange changes in the first three days after sowing were analysed by Q2 technology [28] which can monitor oxygen level changes every few minutes in an enclosed container containing a germinating seed. At the beginning of germination, only respiration occurred, consuming oxygen, which was before any chlorophyll was synthesized. After chlorophyll synthesis and photosynthesis commenced, oxygen was produced, which marked the initiation of the photosynthesis process. The 'Doongara' × 'Reiziq' hybrid initiated photosynthesis more than 10 h earlier than the parents (Table 1), which was consistent with the earlier expression of photosynthesis the component genes.

**Table 1.** Photosynthesis starting time of 'Doongara' and 'Reiziq' and 'Doogara' × 'Reiziq' (hours after sowing (HAS)).

| Line | Photosynthesis Starting Time (HAS) |
|---|---|
| Doongara | $47.49 \pm 1.39$ |
| Reiziq | $49.51 \pm 5.51$ |
| MPV | $48.5 \pm 2.66$ |
| F1 | $35.25 \pm 3.01$ |

Table 1 shows the time-point when the photosynthesis rate reached the compensation point (photosynthesis rate equals respiration rate) after germination.

*3.4. Dark Germination Eliminated Hybrid Vigour*

To further test the impact of photosynthesis on the early development of plants, two groups of 'Doongara' and 'Reiziq' parental plants and their F1 hybrid plants were separately germinated under light and dark conditions. At 14 DAS, the net increase in biomass was calculated in each group (Figure 7). In the group germinated in the light (Figure 2a), the F1 hybrids showed significantly greater biomass at 14 DAS than the mid-parent value (MPV) (*t*-test *p* value = $1.6 \times 10^{-7}$). The F1 hybrids in the dark group did not differ significantly in biomass from the MPV (*t*-test *p* value = 0.3699) (Supplementary Figure S2).

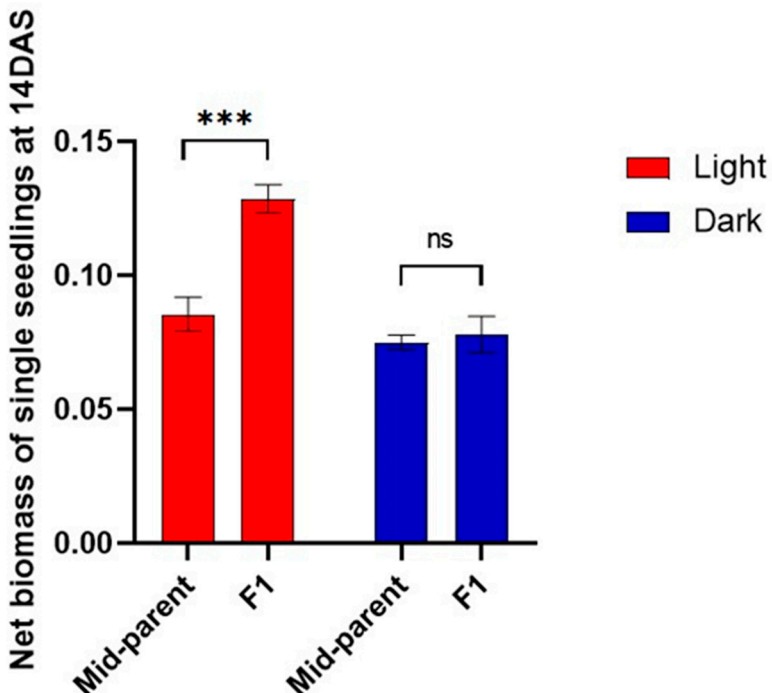

**Figure 7.** Net biomass of single seedling at 14 DAS in light and dark germination groups. *** *p*-value < 0.001; ns—non-significant. *p*-values were calculated using Dunnett's one-way ANOVA. Each plant sample (F1, 'Doongara' and 'Reiziq') has 6 replicates. Error bars represent standard error of the mean (SEM).

Another group of 'Doongara', 'Reiziq' and F1 hybrid plants were grown in darkness for seven days and then transferred to light conditions. In the absence of light, all seedlings developed an etiolated (yellow) growth appearance (Figure S2). After one-day exposure to light, the shoots began to turn green; and the shoots of F1 were greener than the parents. In contrast to the parents, the primary leaves of the F1 hybrid started to open and expand after one day in light (Figure S3). These results demonstrated that 'Doongara' × 'Reiziq' F1 hy-

brid plants switch from skotomorphogenesis (dark development) to photomorphogenesis (light development) earlier than the parental lines.

## 4. Discussion

The Hybrid Mimics, because of their nearly homozygous genomes, produce the F1 hybrid phenotype in each successive generation, generating stable high yielding lines. The F1 hybrid contains genome segments from both parents. Some combinations of genome segments will produce phenocopies of the F1 in morphology, physiology and yield. Under the artificial selection pressure to high grain yield and biomass, Hybrid Mimics have a combination of homozygous alleles from parents. Similarities in the development and altered gene expression between the F1 hybrid and the Mimics relative to the parents, provide an insight into the mechanism of hybrid vigour. Our data indicate that key developmental changes are integral to the generation of hybrid vigour in *japonica* rice. Early germination results in the earlier emergence of each leaf, producing larger leaves which are maintained in subsequent development to produce a higher biomass resulting in increased seed yield.

*Vigour Is Correlated with Early Photosynthesis in Hybrids*

The molecular and cellular mechanisms of hybrid vigour are not fully understood, but the early development of heterosis has been reported in a number of plant species [41,42]. In Arabidopsis, the heterosis of biomass production is established during early seedling development [41], and leaf growth has been shown to be key to biomass heterosis [43]. The transcriptome analysis of young rice seedlings demonstrated that photosynthetic genes have an earlier expression in hybrids compared to parental lines. *Indica* hybrids and hybrid mimics also demonstrated that the increased biomass seen in F1 hybrids is associated with early photosynthetic gene activity [26]. The Q2 analysis of the photosynthesis system provided direct physiological evidence of the earlier commencement of photosynthesis in hybrid *japonica* rice relative to the parental lines.

The dark germination experiment supported the correlation of early photosynthesis with early hybrid vigour. When the plants developed in the dark, normal chloroplasts did not develop from the etioplasts [44] During the seven-day dark development after sowing, F1 and parental seedlings maintained an etiolated state and did not carry out photosynthesis. In the dark, the F1 hybrids did not show seedling vigour compared to parents whereas the hybrids grown in the light had a significantly greater biomass than parental lines. Etiolation in the dark in the absence of photosynthesis abolished the increased vigour of 'Doongara' × 'Reiziq' hybrids at early development stages. These results corroborate the finding in hybrid Chinese cabbage (*Brassica rapa*) with the inhibitor norflurazon [45] where germination for one week on media containing norflurazon abolished hybrid vigour even though two weeks OF further growth in the absence of inhibitor allowed chlorophyll levels to be restored.

When the dark-developing rice seedlings were exposed to light, a series of physiological and biochemical processes began. De-etiolation occurs, leading to photosynthesis [44]. The changes of de-etiolation include the expansion of primary leaves, the synthesis of anthocyanins, and the development of chloroplasts from etioplasts [44]. Once de-etiolation has begun, plants switch to photomorphogenesis [46]. In the *japonica* hybrids, F1 individuals switched to photomorphogenesis more rapidly than parental plants (Figures S2 and S3). One day after exposure to light, the characteristics of de-etiolation, such as primary leaf expansion and chloroplast development (greening) could be observed in the F1 line but not in parental lines. These observations support the contention that early photosynthesis in the germinating seedling of the hybrid is important for the accumulation of seedling biomass in hybrid vigour.

Heterosis in the physiological characteristics of rice has been reported in seedling growth [47], late vegetative growth [48,49], and in the reproductive stages [49]. Our data show that the hybrid vigour of *japonica* rice is initiated at the seedling stage and continues in

later stages. Understanding the initiation of heterosis should help to explain the molecular mechanism of hybrid vigour. The de-etiolation observation indicated that F1 hybrids have earlier readiness for photosynthetic activities than their parents. The dark germination results demonstrated the importance of early photosynthesis for vigour at the seedling stages. The Q2 gas exchange and transcriptome results showed that F1 hybrids begin photosynthesis earlier than parental lines. Hybrid vigour with the rapid accumulation of seedling biomass and leaf area could establish a higher crop growth rate overall [50]. Higher early biomass accumulation enables the reduction in the growth duration of rice varieties, reducing crop water use and providing more flexible cropping rotations [37,51,52].

- *Hybrid Mimics Could Result in a New Japonica Hybrid Breeding System*

Hybrid *indica* rice lines were introduced into Chinese agriculture in 1974. The subsequent 50 years saw the development of hybrid rice which enhanced commercial rice production by approximately two-fold [18,21] and the grain yields of super hybrid rice lines increased to more than 13 t/hm$^2$ in 2012 [53]. Hybrid *indica* rice varieties are also widely grown in the southern USA [54]. Although hybrid varieties have high grain yields, hybrid rice has high grain production costs because of the need for a male sterile female line. There are currently no *japonica* hybrids grown commercially in Australia. Regular hybrid rice breeding in *japonica* rice would require the establishment of a male sterility and fertility restoration system in elite genetic backgrounds.

These requirements do not apply to Hybrid Mimics which present an alternative method to the development of high yielding hybrid varieties in a *japonica* rice production system. The Hybrid Mimic lines, developed by repeated selfing and recurrent selection from selected hand-made F1 hybrids, have F1-like phenotypes and high yields; the lines are stable over subsequent generations, avoiding the restriction of the F1 yield advantage to a single generation. The selection of 'Doongara' × 'Reiziq' Mimics used the F1 hybrid as a reference standard. Plants with equivalent grain yield and biomass to F1 plants were used to generate the subsequent generations; the F5 generation met the criteria for hybrid mimics. Instead of the need to develop a three-line *japonica* male-sterile system, the development of Hybrid Mimic lines of *japonica* could provide the basis of a new breeding system for high yielding rice. Mimic systems could be used to develop high yielding varieties for both *japonica* and *indica* varieties.

**Supplementary Materials:** The following supporting information can be downloaded at: https://www.mdpi.com/article/10.3390/agronomy12071559/s1, Figure S1: The early growth patterns of the other Australian hybrid lines not examined in detail in this study; Figure S2: The etiolated seedlings after seven-day dark germination; Figure S3: The de-etiolated seedlings germinating in the dark followed by a one-day exposure to light; Figure S4: Plant height, panicle number and seed setting rate from parental lines to the F5 generation; Table S1: Partial agronomic measurement of 50 'Doongara' × 'Reiziq' F2 samples; Table S2: Partial agronomic measurement of 50 'Doongara' × 'Reiziq' F3 samples; Table S3: Partial agronomic measurement of 50 'Doongara' × 'Reiziq' F4 samples; Table S4: The primers of target genes and the housekeeper gene for RT-qPCR.

**Author Contributions:** Conceptualization, X.W., W.J.P. and E.S.D.; methodology, Y.Z. and B.O.; software, Y.Z. and B.O.; validation, Y.Z. and B.O.; formal analysis, Y.Z. and B.O.; investigation, Y.Z., B.O. and Y.H.; resources, B.O., Y.H. and W.Y.; data curation, Y.Z.; writing—original draft preparation, Y.Z.; writing—review and editing, B.O., W.J.P. and E.S.D.; visualization, Y.Z.; supervision, X.W., W.J.P. and E.S.D.; project administration, W.J.P. and E.S.D.; funding acquisition, X.W., W.J.P. and E.S.D. All authors have read and agreed to the published version of the manuscript.

**Funding:** This research was funded by UTS International Research Scholarship and UTS President's Scholarship (Appliction No. 196246), the National Key Research and Development Program of China (Grant No. 2016YFD0100406) and the Key Research and Development Program of Sichuan (Grant No. 2021YFYZ0016).

**Institutional Review Board Statement:** Not applicable.

**Informed Consent Statement:** Not applicable.

**Data Availability Statement:** Not applicable.

**Acknowledgments:** We thank Limin Wu, Kylie Elliott and Anyu Zhu for assistance in rice planting.

**Conflicts of Interest:** The authors declare no conflict of interest.

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
