# Peer review of "Hybrid Vigour and Hybrid Mimics in Japonica Rice"

_agronomy, doi:10.3390/agronomy12071559_

Round 1

Reviewer 1 Report

The manuscript provides deeper insight to basic nature of hybrid vigor after crossing rice cultivars with phylogenetic distance. Heterosis of F1 plants were expressed in seedling stage and seed yield. As a new approach hybrid mimic selection was also performed in the subsequent generations. The authors provide data about the role of photosynthesis in hybrid vigor.

Further comments:

1.     It is poorly discussed how hybrid mimic can result in similar yield as F1 plants. There is no control combination where similar selection carried out in the parental population.

2.     There is no explanation for the lack a of MPV in expression of photosynthetic genes on the second day of germination.

3.     In Table 1. the F1 plant show very low photosynthesis. How?

Reviewer 2 Report

The work is devoted to obtaining and researching highly productive hybrid and hybrid Mimics in rice. This is a very important work to maintain food security around the world.

However, there are comments.

In the materials and methods in the part devoted to expression, specify the method by which the calculations were performed (according to Livak or others). It is also necessary to introduce a section dedicated to statistical methods.

In the manuscript, the F5 Mimics line score is given only for grain yield and plant biomass. For the other growth and physiological parameters, only parental lines and F1 data are given. It is not right. Moreover, the expression values are given only for the F1 hybrid.

It also remains unclear how phenotypically similar F1 and Mimics line are.
